# Improving the Safety and Continuity Of Medicines management at Transitions of care (ISCOMAT): protocol for a process evaluation of a cluster randomised control trial

Catherine Powell ,[1,2] Liz Breen ,[1,2,3] Beth Fylan ,[1,2,3] Hanif Ismail ,[1,2] Sarah L Alderson ,[4] Chris P Gale ,[5,6,7] Peter Gardner ,[1,2] Amanda J Farrin ,[8] David P Alldred ,[2,3,9] ISCOMAT Programme Management Team

For numbered affiliations see end of article.

**Correspondence to**
Dr Catherine Powell;
C.Powell2@bradford.ac.uk

## ABSTRACT

**Introduction** A key priority for the UK National Health Service and patients is to ensure that medicines are used safely and effectively. However, medication changes are not always optimally communicated and implemented when patients transfer from hospital into community settings. Heart failure is a common reason for admission to hospital. Patients with heart failure have a high burden of morbidity, mortality and complex pharmacotherapeutic regimens. The Improving the Safety and Continuity Of Medicines management at Transitions of care programme comprises a cluster randomised controlled trial which will test the effectiveness of a complex behavioural intervention aimed at improving medications management at the interface between hospitals discharge and community care. We will conduct a rigorous process evaluation to inform interpretation of the trial findings, inform implementation of the intervention on a wider scale and aid dissemination of the intervention.

**Methods and analysis** The process evaluation will be conducted in six purposively selected intervention sites (ie, hospital trusts and associated community pharmacies) using a mixed-methods design. Fidelity and barriers/enablers of implementation of the Medicines at Transitions Intervention (MaTI) will be explored using observation, interviews (20 patients, 40 healthcare professionals), surveys and routine trial data collection on adherence to MaTI. A parallel mixed analysis will be applied. Qualitative data will be thematically analysed using Framework analysis and survey data will be analysed descriptively. Data will be synthesised, triangulated and mapped to the Consolidated Framework for Implementation Research where appropriate. The process evaluation commenced on June 2018 and is due to end on February 2021.

**Ethics and dissemination** Approved by Research Ethics Committee and the UK Health Research Authority REC: 18/YH/0017/IRAS: 231 431. Findings

### Strengths and limitations of this study

► An evaluation of a cluster randomised controlled trial of a complex behavioural intervention for a high priority healthcare issue.
► Views of patients and health professionals working across both primary and secondary care.
► A key strength is our mixed methods and the flexibility of our approach in light of emerging issues from the study and the shifts in healthcare practice and policy.
► The process evaluation is limited to six purposively selected intervention sites.

will be disseminated via academic and policy conferences, peer-reviewed publications and social media.

**Trial registration number** ISRCTN66212970.

## INTRODUCTION

Heart failure presents a major challenge to healthcare systems globally.[1] In the UK, it is thought to affect the lives of over 920 000 people.[2] In a study of four million individuals, heart failure cases have been found to increase 12% from 2002 to 2014.[3] Myocardial infarction has been identified as a risk for heart failure.[4] Heart failure is treated by a combination of medication, lifestyle changes and interventional surgery depending on the severity of the condition. Typically, the pharmacological treatment pathway involves a combination of medicines that are titrated to the optimal level that patients can tolerate. If not managed well (or if the patient has not been diagnosed and treated in primary care) they may be admitted to hospital for treatment and to be stabilised. Approximately

5% of all emergency hospital admissions are for heart failure.[5] Ongoing treatment plans following discharge from hospital may not be fully implemented due to failures in communication between healthcare providers or a lack of specialist staff in the community.[6 7] Patients may not understand their medicines, what they are for and why they need to take them because, for example, medical language may create a gap in patients' understanding.[8] Patients may therefore deteriorate, which can result in a readmission to hospital or death.[9]

### The Improving the Safety and Continuity Of Medicines management at Transitions of care study

Improving the Safety and Continuity Of Medicines management at Transitions of care (ISCOMAT) is a 5-year National Institute for Health Research (NIHR)-funded research programme, which aims to optimise the way heart failure patients are supported with their medicines when they move from hospital to home. This may contribute to improving patients' health through helping them better understand and use their medicines. ISCOMAT also aims to improve the way health professionals work together in order to improve communication and optimise medicines use when patients return home. Similar international studies have examined patient-centred care transitions for patients hospitalised for heart failure and found clinical outcomes did not improve. ISCOMAT is designed for the UK health system and may show improvements in clinical outcomes in this setting.[10] In earlier work packages, we explored the resilience of the medicines management pathway for heart failure and then used Experienced Based Co-Design to develop the Medicines at Transitions Intervention (MaTI) with patients and professionals.[8 11] The MaTI was designed to make best use of medicines and reduce harm through: the provision of information to the patient; enhanced communication between hospital and the patient's community pharmacist; and increased engagement of the community pharmacist post discharge. We have provided limited information about the intervention here to avoid potential contamination of the ongoing trial. A more detailed description of the MaTI intervention will be published alongside feasibility testing. The cluster randomised controlled trial is testing the effectiveness of the MaTI. Patients will be recruited from cardiology wards in 42 acute National Health Service trusts across England, over approximately 12 months. The recruitment target is 50 patients from each cardiology ward (target n=2100 patients in total). Sites are randomised to either treatment as usual or to the MaTI. A site coordinator is responsible for organising the trial and implementation of the MaTI at each site.

### The process evaluation

Process evaluations are increasingly carried out alongside randomised controlled trials.[12 13] They are particularly well suited to trials of complex interventions in multiple sites where the intervention may be implemented differently throughout sites and help us to understand more about the practical problems encountered and how they were resolved. We will follow the Medical Research Council (MRC) recommendations and guidance on process evaluations.[13] The process evaluation will examine the effect of organisational context and setting on intervention delivery, and how the intervention is best implemented. There are no predefined methods that a process evaluation must adopt although they typically involve mixed methods.[14 15] Although the process and outcomes evaluation will be conducted by separate teams, the trial and process evaluation teams will meet regularly to have oversight.

### Aims and objectives

The aims of the process evaluation are to inform interpretation of the trial findings, inform implementation of the intervention on a wider scale (eg, other long-term conditions) and aid potential future implementation of the intervention.

Our objectives at each of the six process evaluation sites are to:

- ► Determine the degree to which the intervention is delivered (internal fidelity).
- ► Explore and explain the relationship between intervention implementation and the trial outcomes.
- ► Identify barriers and facilitators for the successful implementation and roll out of the intervention (should the intervention be effective).

### METHODS
### Study design

In addition to the aforementioned MRC guidance,[13] we will draw on other relevant literature,[16 17] in particular the Consolidated Framework for Implementation Research (CFIR).[18] The CFIR is a conceptual framework that was developed to guide systematic assessment of multilevel implementation contexts to identify factors that might influence intervention implementation and effectiveness.[19] The MaTI intervention is implemented at multiple levels at each site including trusts, secondary and primary healthcare professionals and patients. The CFIR framework generates a knowledge base for implementation across multiple settings within five major domains: intervention characteristics, inner setting, outer setting, individual characteristics and implementation process. Domains will be explored and additional relevant theoretical frameworks applied as appropriate. We will also draw on human resource management evaluation, specifically the ability, motivation and opportunity (AMO) model,[20 21] and capability, opportunity and motivation (COM-B) model.[22] These models are complementary and highlight how policy interventions that require changes in staff behaviour are shaped by their ability to work in different ways, in terms of skills, their level of motivation and their opportunities to change practice.

The study design is a parallel mixed synthesis study using quantitative and qualitative data from six intervention sites of the total 42 sites in the cluster randomised

controlled trial. Methods will involve non-participant observations, semi-structured interviews and surveys. In order to capture data on barriers and facilitators to implementation we will collect data from health professionals working along the patient pathway (from both secondary and primary care), patients and community pharmacists. We will also use fidelity data on adherence to MaTI that is being collected within each intervention site in the trial. We will explore and explain the relationship between intervention implementation and the trial outcomes in the six process evaluation sites through analysing secondary outcome data. These data will indicate whether the intervention improves patient understanding of their medicines and satisfaction with medicines-related care at 2 and 6 weeks postdischarge and twelve months postregistration from the Patient Experience Survey (PES) (a validated item from Coleman *et al*'s transition measure[23]), alongside observations and implementation data collected as part of the process evaluation. Process evaluations need to be flexible in the context of the ongoing trials they are evaluating and we will record and describe adaptations to this protocol when the findings are published.

### Sampling and recruitment

We aim to purposively sample six intervention sites based on the following criteria:

▶ University and non-university hospitals.
▶ Method for transferring medicines discharge information to community pharmacists' for example, electronic system, post, telephone.
▶ Sites located across different geographic areas of England.

Within each of the six selected sites we will recruit patients and staff for interviews and surveys. We will also conduct observations at each of these sites. The sampling approach will be iterative and following sampling of one-to-two pilot sites; consideration of the initial data in relation to fidelity will be made.

### Patient interviews

Twenty patients in total will be sampled for interviews (three to four patients across each of the six sites). We will adopt a purposive sampling strategy to meet our target characteristics in terms of gender, age, length of diagnosis and fidelity to intervention. After piloting, we may need to modify our approach to enable diversity in relation to our sampling criteria. Patients approached will be those that have participated in the trial and received the MaTI intervention.

### Observations of intervention wards

We will seek permission to conduct non-participant observations of staff at ward level (two and a half hours per site). Staff members will be provided with information sheets and a script for informing patients of the researchers' presence. A maximum of two researchers will be present at any one time. Staff will have the opportunity to opt out of the observation. A potential limitation of conducting overt observation is the Hawthorne effect by which individuals being observed may alter their behaviour because they are aware that they are being studied.[24] We will seek to minimise the impact by informing staff that they are being observed and we wish them to behave as normal.[25]

### Hospital staff interviews and surveys

Thirty hospital staff across six sites will be interviewed. Staff members will include those involved in delivering MaTI, such as heart failure specialist nurses, ward pharmacists, pharmacy technicians, cardiology ward nurses and site coordinators. We will identify and recruit secondary care staff through a combination of two approaches: (1) we will use information obtained during our ward observations about staff roles in delivering the intervention and approach staff directly and (2) we will also liaise with the appointed site coordinator to identify staff that are directly involved in the delivery of MaTI.

We will recruit hospital staff for surveys, aiming to recruit as many staff as possible involved in delivering MaTI. We will identify hospital staff with assistance from the site coordinators and seek informed consent from all staff.

### Community pharmacist interviews and surveys

We will identify and recruit ten community pharmacists for interviews. Community pharmacists contacted through nominated pharmacies in each of the six sites will be interviewed. We will identity through the following two methods: the patient checklist forms (collected by the trials unit) containing details of the pharmacy used by patients in the trial; directly contacting and inviting community pharmacists to participate.

Surveys will be undertaken with community pharmacists involved in delivering the intervention. The number of surveys returned will be dependent on the number of differing community pharmacies patients have used as well as return rates. We will identify community pharmacists for surveys through the methods highlighted above for interviews.

### Community heart failure nurse surveys

We will recruit up to five community heart failure nurses in each of the six evaluation clusters to complete surveys. A maximum of 30 surveys will be conducted in total. Community heart failure nurses will be identified through referrals made to the community heart failure services from the hospitals.

For all interviews and surveys, participants will be provided with an information sheet and informed, written consent will be obtained.

### Data collection
### Patient interviews

Semi-structured interviews with patients will be undertaken in patients' homes 3 months postregistration into the trial and will last approximately 45 min. Interviews, with patients and staff, will be audio recorded and transcribed verbatim. (This protocol was agreed prior to

COVID-19. Data collection methods are being altered in response to the pandemic. For example, recent interviews are being conducted via phone. Our flexible approach to the protocol has facilitated these changes. Changes in methods will be tracked and reported as per pragmatic study development). In the first site, patient data will be collected in a block of three to four relatively early in the implementation of the intervention (in the first 3 months). In subsequent sites, we may sample during middle and late phases of implementation which will be determined based on the data collected. As the trial progresses, we will become more familiar with how the intervention is being implemented. Interview schedules will include a range of questions, probes and prompts and will explore patient experiences of the intervention components.

### Observations of intervention wards

Two researchers (CP/HI) will conduct unstructured and structured observations of clinical staff (heart failure nurses, cardiology nurses, cardiology pharmacists) of adherence to the intervention (content, coverage, frequency and duration). These observations will take place in hospital 6 months postregistration of the first patient recruited at that site, focusing in-depth on the discharge process and introduction of the MaTI toolkit. Observations will be two and a half hours per site designed to capture not only delivery of the MaTI (structured) but also to augment our understanding of the hospital ward culture and environment (unstructured). Focused and general observation data will be collected through the use of a designed structured observation tool and field notes.

The focus of the observations will be on the interaction between staff and patients, particularly around the discharge process and completion/use of the toolkit. Other aspects of the intervention for example, transfer of information, time spent with patients and level of patient understanding will be observed if possible. We will collect quantitative data using structured observations to record actions or behaviours, for example, how information was introduced and provided to patients. The structured observation may pose a greater challenge to collect as it will not be possible to identify in advance when MaTI related activities, such as the introduction of the toolkit, will occur. The two-and-a-half-hour period of observation will be discontinuous. We will wait at the site until relevant activity occurs, and we will liaise with staff within the site to help us identify suitable times to conduct this structured observation. Staff will help us identify when the intervention-related activity is likely to occur. In the unstructured observations, we will seek to develop a more general understanding of the ward culture and the ways the staff interact with patients and each other in the delivery of care. Unstructured data will be collated through field notes.

### Hospital staff interviews and surveys

Semi-structured interviews with hospital staff will be conducted using an interview schedule covering staff experiences in delivering the intervention. These will take place at the hospital and at the end of site trial implementation, that is, 2 weeks postdischarge of the last recruited patient to avoid influencing intervention implementation during the trial. Survey data will be collected through paper questionnaires provided to staff when visiting the hospital at the end of site implementation. A maximum 180 surveys will be conducted.

### Community pharmacist interviews and surveys

Community pharmacy semi-structured interviews will focus on how the pharmacists interacted with the enhanced communication from the hospital and how they engaged with patients post discharge. An interview schedule will be used to collect data on the interventions perceived usability and impact. These will take place at the end of site implementation, that is, 2 months postdischarge of the last recruited patient. Interviews may take place face to face or via phone. Survey data will be collected through questionnaires to staff via post once the intervention has finished at that site. A maximum 300 surveys will be conducted.

### Community heart failure nurse surveys

Community heart failure nurses will be invited to complete postal surveys at the end of site implementation. A maximum 30 surveys will be conducted. We do not plan to interview community heart failure nurses because our primary focus is those delivering the intervention that is, hospital staff and community pharmacists. However, the survey data will explore whether nurses used the toolkit with patients in the community, whether the toolkit enhanced patient care and whether nurses would advocate the use of the toolkit to support patient treatment. Table 1 illustrates the timing of all data collection.

### Analysis

The qualitative analysis will be undertaken using a two-step process:

### Step 1: framework analysis

The process of interpreting the transcripts will take place while interviews are still being conducted. This will give the research team the opportunity to explore emerging themes in detail in subsequent interviews. The interviews and unstructured observations will be analysed using the Framework approach, which involves detailed familiarisation with the data, identifying key themes, interpreting the findings within the context of similar research studies, and considering policy and practice.[26] The emerging analysis will be thematic and iterative with regular discussions taking place with the process evaluation team. This involvement will support our interpretation of the interview data. The analysis will be theoretically informed by COM-B[22] and AMO.[20–22]

**Table 1** Data collection across six process evaluation sites*

| | Interview Patient | Interview Hospital staff | Interview community pharmacists | Focused and general observations | Survey hospital staff | Survey community pharmacists | Survey community Heart failure nurses |
|---|---|---|---|---|---|---|---|
| | 20 in total | 30 in total | 10 in total | 2.5 hours | 180 max. Sampling frame is all hospital staff involved in delivery of the intervention | 300 max. Sampling frame is all community pharmacists that have delivered the intervention | 30 max. Sampling frame is all community heart failure nurses associated with each site |
| | 3 months postpatient registration into trial | 2 weeks postdischarge last patient | 2 months postdischarge last patient | 6 months postimplementation | 2 weeks postdischarge last patient | 2 months postdischarge last patient | 2 months postdischarge last patient |

*Within each site, data collection will be sequential. Across the six sites, data collection will be non-sequential as it will depend on recruitment rates at each site.

## Step 2: CFIR

Following initial theme generation, we will review the data using the CFIR.[18] The analysis will involve mapping barriers and facilitators onto domains within CFIR.

Quantitative data will include survey, structured observations and data on adherence to the intervention. The survey data will be entered into secure databases. Descriptive statistics will be employed to analyse survey data. Data relating to adherence to MaTI is being collected from all intervention sites. This consists of checklists detailing which components of the intervention were implemented for each patient. We will use these data to inform and explain the findings in the process evaluation sites.

### Additional data

Where appropriate, we may consider using the additional data (collected for all sites as part of the wider trial) to inform, explain and triangulate findings with the process evaluation. For example, we may decide that we need more structured information on completion of the MaTI checklist to consider alongside our staff interviews and so clarify which steps of the MaTI different staff members carried out.

Additional data sources include:

▶ Site Feasibility Questionnaire (initial questions sent to sites to assess their suitability to take part in the trial covering areas such as clinical pathways, staffing levels/ types of specialist staff, communication with community pharmacy and the number of patients).
▶ PES (completed by patients at 2 weeks/6 weeks/12 months postdischarge).
▶ MaTI checklist completed in the hospital (monitors adherence to the main components of the intervention).
▶ National Heart Failure Audit Data[27] (reports on the characteristics of patients admitted with acute or subacute heart failure, in-hospital investigations and care, treatment given and the discharge planning and follow-up).
▶ Community pharmacy data collection form (describes implementation within community pharmacy).

Analysis will be integrative in order to clarify and explain the predominant systems and their implications; qualitative and quantitative data will be consolidated through a process of 'parallel mixed analysis'.[28] This includes an independent analysis of the quantitative and qualitative data to provide an understanding of key phenomena and the two understandings will be integrated using meta-inferences. For example, key findings will be generated iteratively, explicitly supported by quantitative data (such as structured observational and survey data) and substantiated or augmented by thematic qualitative data (such as interview data and field notes) and vice-versa. Survey data will be compared across the six clusters to identify any differences in staff perceptions of the barriers and facilitators to delivery. Analysis will be descriptive.

The combined analysis will, therefore, meet our key aims to inform interpretation of the trial findings, inform

implementation of the intervention on a wider scale (eg, other long-term conditions) and aid potential future implementation of the intervention. This will be achieved by providing an in depth understanding of the overall implementation, mechanisms of impact and external factors (infra structure) that may influence delivery and functioning of the intervention.

## Patient and public involvement
The ISCOMAT study has a patient-led steering group that is involved in all stages of the research process, including the process evaluation. Due to the process evaluation's iterative design we will regularly consult with the group via meetings and phone/email. The group will continue to be consulted on the research design, questions, outcome measures and findings. In particular the members of the group contribute to reviews and evaluations, as well as reading and considering study and consultation documents from a patient perspective. The group's expertise through their experiences of living with heart failure will be crucial in understanding patients' experiences with the MaTI intervention in particular.

## ETHICS AND DISSEMINATION
The process evaluation has been approved as part of the ISCOMAT trial by the Research Ethics Committee and the UK Health Research Authority REC: 18/YH/0017/ IRAS: 231 431.

Findings will be disseminated via academic and policy conferences, peer-reviewed publications, social media, for example, Twitter, with further avenues for dissemination to be agreed on with our patient led steering group.

## CONCLUSION
In this paper, we have described the design and methods for the mixed-methods process evaluation of the NIHR-funded ISCOMAT cluster randomised controlled trial which will test the effectiveness of a complex behavioural intervention aimed at improving medications management at the interface between hospital and community for patients with hospitalised with heart failure. This process evaluation protocol demonstrates the importance of process evaluations for understanding outcomes in the clinical trial, as well as providing guidance for future process evaluations. We have followed the MRC recommendations and guidance on the delivery of process evaluation[13] in order to support the standardisation of process evaluations.

## Author affiliations
[1]School of Pharmacy and Medical Sciences, Faculty of Life Sciences, University of Bradford, Bradford, UK
[2]Wolfson Centre for Applied Health Research, Bradford, UK
[3]Bradford Institute for Health Research, NIHR Yorkshire and Humber Patient Safety Translational Research Centre, Bradford, UK
[4]Leeds Institute of Health Sciences, University of Leeds, Leeds, UK
[5]Leeds Institute of Cardiovascular and Metabolic Medicine, University of Leeds, Leeds, UK
[6]Leeds Institute for Data Analytics, University of Leeds, Leeds, UK
[7]Department of Cardiology, Leeds Teaching Hospitals NHS Trust, Leeds, UK
[8]Clinical Trials Research Unit, University of Leeds, Leeds, UK
[9]School of Healthcare, University of Leeds, Leeds, UK

**Acknowledgements** We would like to thank Alison Blenkinsopp, Gerry Armitage, Lauren Moureau, Jan Speechley, the ISCOMAT Patient-led Steering Group, the ISCOMAT Trial Management Group, Trial Steering Committee and the Programme Steering Committee.

**Collaborators** Our collaborators include members of the wider ISCOMAT Programme Management Team who contributed to previous work packages and the ongoing programme including: Jon Silcock, David K. Raynor, Robert Turner, John Wright, Ian Kellar, Roberta Longo, Ivana Holloway, Chris Bojke, Leeds Clinical Trials Research Unit.

**Contributors** CP, LB, BF, HI, SA, CG, PG, AF and DPA developed the detail of the process evaluation protocol. CP drafted the manuscript and all authors reviewed it critically for intellectual content and approved the final version submitted for publication.

**Funding** This study is funded by the National Institute for Health Research (NIHR) (Programme Grants for Applied Research (Grant Reference Number RP-PG-0514-20009)). This research was supported by the National Institute for Health Research (NIHR) Yorkshire and Humber Patient Safety Translational Research Centre (NIHR Yorkshire and Humber PSTRC).

**Disclaimer** The views expressed are those of the authors and not necessarily those of the NIHR or the Department of Health and Social Care.

**Competing interests** None declared.

**Patient and public involvement** Patients and/or the public were involved in the design, or conduct, or reporting, or dissemination plans of this research. Refer to the Methods section for further details.

**Patient consent for publication** Not required.

**Provenance and peer review** Not commissioned; externally peer reviewed.

**ORCID iDs**
Catherine Powell http://orcid.org/0000-0001-7590-0247
Liz Breen http://orcid.org/0000-0001-5204-1187
Beth Fylan http://orcid.org/0000-0003-0599-4537
Hanif Ismail http://orcid.org/0000-0002-7885-6648
Sarah L Alderson http://orcid.org/0000-0002-5418-0495
Chris P Gale http://orcid.org/0000-0003-4732-382X
Peter Gardner http://orcid.org/0000-0002-8799-0443
Amanda J Farrin http://orcid.org/0000-0002-2876-0584
David P Alldred http://orcid.org/0000-0002-2525-4854

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
