## [Reviewer comments · BMJ Open]

ARTICLE DETAILS

TITLE (PROVISIONAL)	Improving the Safety and Continuity Of Medicines management at Transitions of care (ISCOMAT): protocol for a process evaluation of a cluster randomised control trial
AUTHORS	Powell, Catherine; Breen, Liz; Fylan, Beth; Ismail, Hanif; Alderson, Sarah; Gale, Chris; Gardner, Peter; Farrin, Amanda; Alldred, David

VERSION 1 – REVIEW

REVIEWER	Laressa Bethishou Chapman University School of Pharmacy, Irvine CA, USA
REVIEW RETURNED	18-Aug-2020

GENERAL COMMENTS	This is great, as a TOC pharmacist in the US, I'd love to learn more about this model and how it may be adapted to our health system.
---

REVIEWER	Yogini Jani University College
REVIEW RETURNED	21-Sep-2020

GENERAL COMMENTS	This protocol for process evaluation of an RCT aims to test the effectiveness of a complex behavioural intervention to improve medicines management at transitions of care using the use of mixed methods across 6 sites and consolidated framework for implementation research has been ongoing since June 2018. Overall, a clearly formed and written protocol, using appropriate frameworks and models to guide the methods as well as analysis. The following comments need further clarification or detail. Introduction page 6, line 16 - make clear that recruitment target is 50 patients at each of the 42 sites. Methods page 7, line 30 - make clear that this is a sub-selection of 6 intervention sites from the total 42 sites in the RCT. Methods page 8, 9 observations - not clear what exactly will be observed; is this a continuous 2.5 hours? how will that work in practice if there is no intervention related activity during the observation period? Is 2.5 hours per site long enough to gain any/ meaningful insights? Methods page 8/9, is there an upper limit to the number of surveys that will be conducted?
--

	Methods page 9, line 24 - state the total number of nurses to be surveyed; why are nurses not being interviewed? Data collection page 9, line 38 - have any adaptations been made due to COVID-19 e.g. telephone interviews
--	---

VERSION 1 – AUTHOR RESPONSE

Reviewer 1 comments	Actions/comments
This is great, as a TOC pharmacist in the US, I'd love to learn more about this model and how it may be adapted to our health system.	Thank you, we are delighted about your interest in the ISCOMAT study. (No further actions taken in the article in response to reviewer 1 comments).

Reviewer 2 comments	Actions/comments
This protocol for process evaluation of an RCT aims to test the effectiveness of a complex behavioural intervention to improve medicines management at transitions of care using the use of mixed methods across 6 sites and consolidated framework for implementation research has been ongoing since June 2018. Overall, a clearly formed and written protocol, using appropriate frameworks and models to guide the methods as well as analysis.	Thank you for the careful reading of our paper. We have clarified the procedures by answering the queries below and making the necessary amendments.
Introduction page 6, line 16 - make clear that recruitment target is 50 patients at each of the 42 sites.	Introduction text (now page 4) has been changed to the following; 'Patients will be recruited from cardiology wards in 42 acute NHS trusts across England, over approximately 12 months. The recruitment target is 50 patients from each cardiology ward (target n=2100 patients in total).'
Methods page 7, line 30 - make clear that this is a sub-selection of 6 intervention sites from the total 42 sites in the RCT.	Methods text (now page 5) has been changed to the following; 'The study design is a parallel mixed synthesis study using quantitative and qualitative data from six intervention sites of the total 42 sites in the cluster randomised controlled trial.'
Methods page 8, 9 observations - not clear what exactly will be observed; is this a continuous 2.5 hours? how will that work in practice if there is no intervention related activity during the	Methods text (now page 8) changed to the following; 'The two-and-a-half-hour period of observation will be discontinuous. We will wait at the site until relevant activity occurs, and we will liaise with staff within the site to help us identify suitable times to conduct this

observation period? Is 2.5 hours per site long enough to gain any/ meaningful insights?	structured observation. Staff will help us identify when the intervention related activity is likely to occur.'
Methods page 8/9, is there an upper limit to the number of surveys that will be conducted?	Methods text (now page 8 and 9) and table 1 (now page 9) updated 'A maximum 180 surveys will be conducted.' 'A maximum 300 surveys will be conducted.' 'A maximum 30 surveys will be conducted.' In table '180 max.' '300 max.' '30 max. Sampling frame is all community heart failure nurses associated with each site.'
Methods page 9, line 24 - state the total number of nurses to be surveyed; why are nurses not being interviewed?	Methods text (now page 7 and page 9) changed to the following; 'Community heart failure nurse surveys We will recruit up to five community heart failure nurses in each of the six evaluation clusters to complete surveys. A maximum of 30 surveys will be conducted in total.' 'A maximum 30 surveys will be conducted. We do not plan to interview community heart failure nurses because our primary focus is those delivering the intervention i.e. hospital staff and community pharmacists. However, the survey data will explore whether nurses used the toolkit with patients in the community, whether the toolkit enhanced patient care and whether nurses would advocate the use of the toolkit to support patient treatment.'
Data collection page 9, line 38 - have any adaptations been made due to COVID-19 e.g. telephone interviews	Data collection text (now page 7) changed to the following; '(This protocol was agreed prior to Covid-19. Data collection methods are being altered in response to the pandemic. For example, recent interviews are being conducted via phone. Our flexible approach to the protocol has facilitated these changes. Changes in methods will be tracked and reported as per pragmatic study development).'

VERSION 2 – REVIEW

REVIEWER	Yogini Jani University College London Hospitals NHS Foundation Trust, England, United Kingdom
REVIEW RETURNED	29-Oct-2020
GENERAL COMMENTS	All comments have been addressed satisfactorily.